# Obliviate! Reviewing Neural Fundamentals of Intentional Forgetting from a Meta-Analytic Perspective

**DOI:** 10.3390/biomedicines10071555

**Published:** 2022-06-29

**Authors:** Olga Lucia Gamboa, Hu Chuan-Peng, Christian E. Salas, Kenneth S. L. Yuen

**Affiliations:** 1School of Psychology, University of Sydney, Sydney 2006, Australia; 2EQness, Sydney 2034, Australia; 3School of Psychology, Nanjing Normal University, Nanjing 210024, China; hcp4715@gmail.com; 4Leibniz Institute for Resilience Research, 55122 Mainz, Germany; 5Laboratory of Cognitive and Social Neuroscience, Faculty of Psychology, Diego Portales University, Santiago 7510457, Chile; salasriquelme@gmail.com; 6Neuroimaging Center (NIC), Focus Program Translational Neuroscience, Johannes Gutenberg University Medical Center, 55131 Mainz, Germany

**Keywords:** intentional forgetting, directed forgetting, fMRI, neuroimaging, meta-analysis, Activation Likelihood Estimation (ALE), Latent Dirichlet Allocation (LDA)

## Abstract

Intentional forgetting (IF) is an important adaptive mechanism necessary for correct memory functioning, optimal psychological wellbeing, and appropriate daily performance. Due to its complexity, the neuropsychological processes that give birth to successful intentional forgetting are not yet clearly known. In this study, we used two different meta-analytic algorithms, Activation Likelihood Estimation (ALE) & Latent Dirichlet Allocation (LDA) to quantitatively assess the neural correlates of IF and to evaluate the degree of compatibility between the proposed neurobiological models and the existing brain imaging data. We found that IF involves the interaction of two networks, the main “core regions” consisting of a primarily right-lateralized frontal-parietal circuit that is activated irrespective of the paradigm used and sample characteristics and a second less constrained “supportive network” that involves frontal-hippocampal interactions when IF takes place. Additionally, our results support the validity of the inhibitory or thought suppression hypothesis. The presence of a neural signature of IF that is stable regardless of experimental paradigms is a promising finding that may open new venues for the development of effective clinical interventions.

## 1. Introduction

Forgetting is an important adaptive mechanism essential for correct memory function. It helps regulate the content of memory storage in a way that only appropriate, relevant, and up-to-date information is kept [1,2]. The study of forgetting in animals has taken different forms, from behavioral measures like extinction of conditioned responses, pharmacological manipulations to block memory consolidations, to optogenetic manipulations of engram, or a mixture of all these techniques [3,4,5,6]. In human studies, memory extinction has been extensively studied in the domain of fear memory processes [5,7,8,9,10,11]. In parallel, a good amount of work has been devoted to study the failure of memory retention, i.e., incidental forgetting, and its neural correlates on declarative memory [12,13]. Both arms of study have revealed an overlapping brain network consisting of elevated activities in the ventromedial prefrontal cortex, anterior cingulate cortex, precuneus, coupled with the down regulation of the hippocampus, to support forgetting processes [7,8,12,13].

Incidental forgetting or extinction are both considered automatic processes. Intentional forgetting (IF), by contrast, represents an individual’s active, volitional pursue to get rid of unwanted information [14]. It has its historical root in Freudian theory, known as suppression, and subsequently re-examined in a neurocognitive framework using neuroimaging techniques [15,16,17]. The relevance of intentional or motivated forgetting goes beyond mnemonic processes, as it is key for preserving good psychological health, supporting emotion regulation, structuring cognition, and facilitating behavioural flexibility [2]. Understanding the processes underlying intentional forgetting is of great value not only for cognitive scientists but for the medical community trying to develop optimized treatments directed to population suffering from disorders related to the inability to regulate intrusive thoughts. This understanding is even more important in face of the replication crisis of other memory manipulation techniques (e.g., memory extinction by reactivation, [18]). As such, we will solely focus on discussing IF in the current review.

### 1.1. Experimental Paradigms

Several experimental paradigms have been developed to study IF. They all follow the same principle: participants first learn some information, that later they will be instructed to either forget or remember [16,19,20]. The main difference between these paradigms lies in whether forgetting occurs at the encoding or retrieval phase.

In the think/no-think (TNT) paradigm, participants first go through a learning phase, studying cue-target pairs of items. In the critical phase (think/no-think task) only cue items are presented followed by an instruction to remember (think condition) or to suppress (no-think condition) the associated target. For the no-think condition, participants are instructed to fully avoid allowing the target to enter conscious awareness. Item pairs that are only shown in the learning phase but not the critical phase serve as the baseline condition. In the test phase, the cues from all three conditions (think, no-think and baseline) are shown and participants are asked to recall the correct target items [16,21]. Items in the no-think condition are recalled worse than in the other two conditions, while items in the think condition are recalled better than the ones in the baseline condition [16]. Essentially, the frequency of no-think operations and successful forgetting follows a dose-response relationship. Behavioural outcomes are explained via two theoretical accounts, the inhibitory hypothesis in which brain mechanisms related to inhibitory control are recruited by the no-think items [16] and the interference/substitution hypothesis suggesting that interference coming from information other than the no-think items, further aids forgetting [22,23].

The list–method directed forgetting paradigm, on the other hand, has a simpler experimental design. In the initial phase, participants are instructed to learn a list of items (list 1) to be tested later. Half of the participants are told to forget list 1 (forget condition), and then a second list of items (list 2) to be learned is presented to all participants. During the test phase, participants in both conditions are asked to recall both lists [24]. Participants in the forget condition perform worse at recalling list one, but recall list 2 better than participants in the remember condition [24]. Behavioural results have been explained through the retrieval inhibition hypothesis, in which the list to be remembered (list 2) interferes with the previously learned list (list 1) impairing its recall [20,25].

Finally, a variation of the list–method paradigm, is the item–method directed forgetting paradigm. Words are presented one by one to the participants, immediately followed by an instruction to either remember (R) or forget (F). In the test phase, participants are asked to recall all words regardless of the given instruction [26], displaying a better capacity to recall items to be remembered than items instructed to be forgotten [26,27]. In light of neuroimaging findings that show intentional forgetting as an active and complex mechanism [28,29,30,31,32], behavioural outcomes can be explained via the attentional inhibition-executive control hypothesis. Here, items to be forgotten experience an active inhibition that will remove them from working memory, limit their access to attentional resources and avoid future activations [33]. Meanwhile, the executive system actively regains processing resources boosting the rehearsal of items to be remembered [34,35,36].

### 1.2. Hypotheses of Brain Mechanisms: Thought Suppression and/or Substitution

Although the experimental paradigms mentioned above can successfully induce forgetting, there is not a clear understanding of the exact neuropsychological processes used to achieve IF. Researchers have also not yet reconciled to a cognitive framework explaining the underlying mechanisms supporting IF, therefore we see conceptually overlapping hypotheses are constantly being proposed. In our previous work we have qualitatively analyzed subjects’ reported strategies employed during an item-method paradigm and found evidence of both active suppression and self-induced interference as predominant strategies to forget intentionally [29]. Similar hypotheses have been put forward by other research groups to explain how intentional forgetting occurs in the brain. First, the inhibitory or thought suppression hypothesis, which refers to a direct suppression of the unwanted memories, and second, the substitution or thought replacement hypothesis, a mechanism in which to-be-forgotten material is replaced by irrelevant content [29,37]. Experimental findings suggest that these two hypothesized processes are subserved by discrete neural circuitries: a fronto-hippocampal circuit that supports thought suppression/inhibition, and the ventral lateral prefrontal cortex (VLPFC) and the inferior frontal gyrus (IFG, labelled as caudal prefrontal cortex cPFC in the original paper) that supports thought substitution/replacement [38]. Attempts to test these two hypotheses have so far produced mixed results, as it has not been possible to replicate the differential patterns of neural activation comparing inhibition and thought substitution/replacement [39]. Therefore, it would be interesting to see if the two hypothesized patterns of neural activation can be observed in a meta-analysis of neuroimaging studies using the different IF paradigms.

### 1.3. Current Meta-Analysis

We have two main goals performing this meta-analysis: (1) to summarize and examine in a quantitative manner the neural correlates of intentional forgetting, (2) to establish to what extent the proposed neurobiological models (thought suppression and/or substitution) are supported by the reported data. We used two different meta-analytic algorithms, Activation Likelihood Estimation (ALE) & Latent Dirichlet Allocation (LDA) to provide complementary analyses on the convergence and divergence of brain activations reported in the literature [40]. The ALE algorithm is conventionally used for coordinate-based meta-analysis of neuroimaging results [41,42]. It identifies areas that exhibit a convergence of reported coordinates across experiments that is higher than expected under a random spatial association. While ALE analysis focuses on the convergence of activities across studies, complementary analysis using the Latent Dirichlet Allocation (LDA) algorithm can look into the divergence of neural circuitry underlying intentional forgetting. LDA is a data-driven Bayesian framework originally designed to perform automatic semantic extraction from a corpus of text. In recent years LDA and its variant, Author Topic Modelling (ATM), have been utilized to analyze neuroimaging data in order to reveal the latent cognitive network across experimental tasks or clinical conditions [40,43,44,45,46]. A combination of ALE and LDA is a novel approach that will strengthen our understanding of the mechanisms underlying intentional forgetting and may yield valuable information that can be very useful during the development of effective treatments for neuropsychiatric disorders related to intrusive thoughts and the inability to detach from unwanted memories.

## 2. Materials and Methods

### 2.1. Literature Search and Article Selection

Following the PRISMA 2009 flow diagram [47], we reported the literature search and the articles selection process as below (see Figure 1A). First, we performed an online-search to identify studies matching our scope in PubMed (date: May 2022), using the following syntax: intentional forgetting [Title/Abstract] OR motivated forgetting [Title/Abstract] OR instructed forgetting [Title/Abstract]) AND ((Magnetic Resonance Imaging) OR Directed forgetting [Title/Abstract]) AND ((Magnetic Resonance Imaging) OR (functional Magnetic Resonance Imaging) OR (Positron emission tomography)) filter English.

### 2.2. Activation Likelihood Estimation (ALE) Analysis

Additional studies were identified through additional database (Google scholar) and the reference list obtained from the screened articles by the author (OLG, KY). After deleting the duplicated items, our search resulted in 147 studies for further screening. All studies were then screened according to our eligibility criteria: below): (1) studies that investigated intentional forgetting using fMRI and PET; (2) studies with healthy participants that were young to mid-aged adults, i.e., aged 18–45 years old. Studies focused on patients but reporting results from a healthy control group were included; (3) studies reporting whole-brain analysis (articles with results derived from only ROI analyses were excluded); (4) studies reporting standard reference frames such as MNI or Talairach; (5) if multiple papers used the same dataset, only one of these papers was included. Details of article selection is presented in Figure 1A. Conceptualization of this meta-analytoc review was pre-registered at the Open Science Foundation (https://osf.io/xaq5k, DOI: 10.17605/OSF.IO/XAQ5K, 2 June 2022).

We used the revised ALE algorithm for the coordinate-based meta-analysis of neuroimaging results [48,49]. This algorithm identifies areas that exhibit a convergence of reported coordinates across experiments that is higher than expected under a random spatial association. To account for the uncertainty associated with each activation cluster, ALE algorithm constructs 3D Gaussian probability distributions of activation likelihood based on each peak voxel. The Full-Width Half-Maximum (FWHM) of these Gaussian functions were determined based on the between-subject variance by the number of examined subjects per study so that foci with larger sample sizes can be modeled by “smaller” Gaussian distributions because they provide more reliable approximations of the “true” activation effect [48].

The probabilities of all foci reported in a given experiment were then combined for each voxel, resulting in a modeled activation (MA) map [41]. Taking the union across these MA maps yielded voxel-wise ALE scores that described the convergence of the results across experiments at each location of the brain. To distinguish “true” convergence among studies from random convergence (i.e., noise), we compared ALE scores to an empirical null distribution reflecting a random spatial association among experiments. Here, a random-effects inference was invoked, focusing on the inference on the above-chance convergence among studies rather than the clustering of foci within a particular study. Computationally, deriving this null-hypothesis involved sampling a voxel at random from each of the MA maps and taking the union of these values in the same manner as performed for the (spatially contingent) voxels in the true analysis, a process that can be solved analytically [41]. The *p*-value of the “true” ALE was then given by the proportion of equal or higher values obtained under the null-distribution. The resulting non-parametric p-values were then thresholded at the *p* < 0.05 (cluster-level corrected for multiple-comparison; cluster-forming threshold *p* < 0.001 at voxel level) [41]. All significant clusters were reported, and the volume, weighted center and locations, and Z-scores at the peaks within the regions are given.

### 2.3. Latent Dirichlet Allocation (LDA) Analysis

While ALE analysis focuses on the convergence of activities across studies, complementary analysis using the Latent Dirichlet Allocation (LDA) algorithm can look into the divergence of neural circuitry underlying intentional forgetting. LDA is a data-driven Bayesian framework originally designed to perform automatic semantic extraction from a corpus of text. In recent years LDA and its variant, Author Topic Modelling (ATM), have been utilized to analyze neuroimaging data in order to reveal the latent cognitive network across experimental tasks or clinical conditions [40,43,44,45,46,50]. In brief, LDA/ATM is data-driven Bayesian framework that estimates the latent cognitive component across observed voxel-wise activations (Figure 1C). As there are only two experimental paradigms available for neuroimaging studies of IF, applying ATM will tend to overfit the data with an extra layer of constrain. As such, LDA, i.e., equivalent to an ATM treating each individual study as having their own author, is more appropriate to model the data. Scripts for running LDA/ATM are based on the following Github despository (https://github.com/ThomasYeoLab/CBIG/tree/master/stable_projects/meta-analysis/Ngo2019_AuthorTopic, accessed on 27 March 2020). Conditional probabilities Pr(Voxel|Factor) and Pr(Factor|Study) are being estimated by the Collapsed Variational Bayesian (CVB) inference algorithm (with alpha = 100, eta = 0.01, 100 random seeds for each K), and model selection determining the number of optimal factors (K) is done by the Bayesian Information Criterion (BIC) (Figure 1D).

## 3. Results

Following standardized procedures, we performed keyword-based literature search, screening, for study inclusion into our meta-analysis (see Methods for details). In brief, we searched for functional neuroimaging studies with healthy participants that performs an IF task up to May 2022. As the study on the neural correlates of IF is relatively new, the majority of studies are conducted on young healthy adults so we focus on samples aged between 18–45 years old. The article selection resulted in 23 studies, with 466 subjects, 159 foci. While it is almost impossible to estimate the number of unpublished null findings that exist, a recent simulation study suggested that a minimum of 20 experiments, in combination with cluster-level correction, should provide adequate power and sensitivity to reveal a robust effect [51]. Our article selection should thus be considered representative of the subject matter. Figure 1A. presents the PRISMA flowchart on study selection (see also Table 1 for a list of the studies included). The ALE results revealed that four brain regions were convergently activated by directed forgetting > remembering contrast: right superior frontal gyrus (rSFG), right inferior parietal lobe (rIPL, including both supramarginal and angular gyri), bilateral middle frontal gyrus (MFG) (see Table 2, Figure 1B). The LDA analysis revealed a one-factor solution as the most optimal model (Figure 1D). The latent cognitive component revealed by this solution is highly similar to the ALE analysis, showing the involvement of bilateral middle frontal gyri, bilateral IPL, plus additional brain networks not revealed by ALE, involving bilateral hippocampal complex, precuneus, bilateral middle cingulum, primary visual cortex and cerebellum (Table 2, Figure 1E).

## 4. Discussion

During this meta-analysis we examined the neuroimaging literature on intentional forgetting, as a means to get a better understanding of brain structures supporting such an important mechanism. With two different methods (ALE and LDA) we tested the convergence and divergence of underlying neural circuitry that supports IF. Comparing the two resultant activation maps, we found strikingly similar patterns of activation foci in right superior frontal gyrus, bilateral middle frontal gyri, and right inferior parietal lobe. Additional brain network consisting of the hippocampal complex and surrounding temporal areas, middle cingulum, precuneus, primary visual cortex, and cerebellum was revealed with LDA. ALE searches for convergence of neural activation hotspots observed across the selected studies, irrespective of the paradigm used and sample characteristics. Therefore, our results from ALE can be considered the core brain areas supporting IF that generalized across experimental paradigms and studies. Alternatively, LDA search for divergent, latent neural activities that varies in individual studies. Results from LDA can be considered a network of co-activated brain regions that varies in activation depending on the task and stage when IF happened.

### 4.1. Core IF Brain Regions

The converging neural clusters, right superior frontal gyrus (rSFG), right inferior parietal lobe (supramarginal gyrus/angular gyrus included), and bilateral middle frontal gyri (rMFG), are shown by both meta-analytic analyses to be correlates of intentional forgetting. Each of these areas has shown to have an active participation in tasks involving attentional control and inhibition. For instance, the right SFG has been associated with inhibitory control guided by “top-down” processes [66] and cognitive update of memory representations [67]. Its engagement in cognitive functions such as memory or attention may be related to its anatomical and functional connections with relevant frontal regions such as MFG [68].

The MFG has been considered to be an important contributor during retrieval processes. According to hemispherical specializations, attentional and response selection mechanisms have been attributed to the left MFG, while monitoring processes have been linked to right MFG activation [69,70]. Additional to this, it is thought that the right MFG leads attentional control processes by reorienting attention from the external to the internal environment [71] and by flexibly adjusting exogenous and endogenous attention according to the task at hand [72]. Notably, studies with patients suffering right frontal lobe injury support the idea of the right frontal regions as key areas during modulation of attentional processes [71,73] and memory retrieval. Particularly in these studies, these patients were unable to regulate rehearsal and retrieval processes [71].

Meanwhile, the right SMG/AG part of the inferior parietal lobe is a brain region instrumental in two of the main components of attention, alertness and focus on a task and attentional shift to respond to novel, salient information [73]. Being part of the ventral posterior parietal cortex (VPC) and of the ventral fronto-parietal attentional system (comprised of the ventral frontal cortex: middle and inferior frontal gyri, the inferior parietal lobe: supramarginal and angular gyri, and the right temporoparietal junction (TPJ), this region is thought to moderate bottom-up attention [72,74]. And interestingly, its degree of activation has been directly linked to encoding failure [56,75,76].

### 4.2. Supportive IF Network

In addition to the core IF brain regions shown by the ALE analysis, LDA further revealed a divergent group of brain areas that co-activate to support IF. These loosely defined network consist of hippocampal complex and surrounding temporal areas, middle cingulum, precuneus, primary visual cortex and cerebellum. Of particular interest among these brain regions is the role of hippocampal complex, including hippocampus and parahippocampal cortices, in intentional forgetting. Depue and colleagues in addition to the task-based activation reported (and included in our meta-analysishad used functional connectivity during the think-no-think paradigm, as well as fractional anisotropy to provide empirical support for the functional and structural connections between rMFG and hippocampus during forgetting processes. They found that functional communication between the rMFG and hippocampus is supported by the integrity of the cingulum bundle. And that increased integrity of the anatomical pathway was a predictor of the functional connectivity between these two regions during intentional forgetting. Finally, they reported that both structural and functional connections mediated behavior, arguing that there is an ongoing elemental interplay between, brain structure, brain function, and behavior [77]. The functional coupling between hippocampus and rMFG was further demonstrated by Schmitz et al. who used Magnetic Resonance Spectroscopy (MRS) to investigate GABAergic neurotransmission in the hippocampus. They showed that GABAergic inhibition predicts functional coupling between rMFG and hippocampus that is enhanced during retrieval suppression in the think-no-think paradigm [78]. These results, together with the idea that during intentional forgetting frontal regions fulfill an important role as a cognitive control system that modulates parietal activity (in charge of attentional processes) [31], and, other brain structures (involved in mnemonic processes) such as the medial temporal lobe (MTL) [31,38], could be strong indicators of the cooperation between attentional and inhibitory systems to support to act of intentionally forgetting.

Other regions part of this “network” such as the temporal gyrus, orbitofrontal gyrus, and cerebellum, to name some, have been less studied in the context of IF probably because of their lack of direct involvement in the inhibition process. However, their implication may be associated with functions subserving forgetting. For example, the orbital frontal gyrus and temporal gyrus are known to interact with parietal and frontal areas to assist attentional switching an important mechanism in IF [79,80]. Similarly, the cerebellum, a region that is undeniably less known for its role in cognitive functions, has been found to be of great importance in attention [81]. In a clinical study, Gottwald et al. (2003), found that patients with cerebellar damage had difficulties performing a shifting-attention task [82]. The fact that the performance was poor but not eliminated, was interpreted as an indication of the role of the cerebellum as a center of preparation and optimization of higher cognitive functions such as attentional processes [54,83], which have been reported to be necessary for successful forgetting [29].

### 4.3. Thought Suppression and/or Substitution?

Benoit et al. [38] hypothesized two neurocognitive processes supporting IF: *direct suppression* mediated by an increase level of neural activation in DLPFC, coupled with attenuated activations in the hippocampus, or *thought substitution* mediated by increased activations in both right IFG (labelled as cPFC in the original paper) and VLPFC. In our current meta-analyses, the observed neural activations (either core or distributed IF brain regions) do coincide with the ROIs specified in Benoit et al.’s hypotheses.

The core IF regions we identified and discussed above, involving the frontal-parietal circuit, is strongly implicated in cognitive inhibitory processes. Additionally, our LDA results showed frontal-hippocampal involvement in IF and this frontal-hippocampal network resembles the direct suppression processes. It is important to note that the frontal-hippocampal network proposed by Benoit et al. was identified by means of functional connectivity analysis, and these findings are not included in the current meta-analysis (Table 1). Therefore, the observed ALE and LDA activation patterns can be treated as independent verification of Beniot et al.’s hypothesis.

Upon careful scrutiny of the more extensive LDA findings and the distributed IF network, we do observe IFG/cPFC and VLPFC involvement during IF (Figure 2). This finding provides some hints for the existence of thought substitution processes during IF. Nevertheless, it should be cautioned that a meta-analysis like the current one has no way to access individual’s strategy used during IF, but only relies on the observed patterns of neural activation to make a reverse inference on the cognitive processes involved. Linking back to the qualitative analysis on the forgetting strategy in our previous study [29], it is very likely that both inhibition/suppression and thought substitution processes co-exist, or even work in tandem to enhance IF. Further experimental studies should test in detail the differential contribution of IF strategy to achieve successful forgetting.

### 4.4. Applicability of the Findings

Research in the field of forgetting has provided relevant knowledge about how (mostly) the (healthy) brain deals with unwanted information. These results are of great importance for psychiatric disorders where individuals are constantly challenged by involuntary intrusions of unwanted memories. Knowing that purposely trying to forget an unwanted memory, triggers a cascade of mechanisms that leads to obstruction of memory representation and limits its future recovery, may be taken into account to develop new strategies that will help maintain unwanted memories out of awareness.

However, experiments have been mainly performed on healthy participants, and as such, results may not be fully applicable to people with disorders involving problematic thoughts (post-traumatic stress disorder, obsessive-compulsive disorder, depression, etc.), since, resistance or suppression of unwanted memories in a related clinical population may have detrimental outcomes for emotional and mental health [84]. For instance, experiments conducted on patients with anxiety, have shown that using suppression as a coping strategy, reduced the involuntary appearance of anxious thoughts. However, this effect was temporary, and a rebound effect was observed 7 days after the experimental session [85].

In general, studies performed in natural settings have shown that a repressive coping strategy significantly favors the appearance of traumatic memories [49,50], indicating that a more adaptive method to manage intrusive disturbing thoughts in a clinical population is to work with them rather than suppress them [52,85]. Yet, it has been reported that, lack of intentional inhibition of unwanted material results in unsuccessful forgetting [29]. This may imply that to appropriately reduce the strength and appearance of intrusive distressing memories, a certain level of intentional suppression is required, besides the regulation of their cognitive and emotional response [17]. Thus, the formulation of new strategies to regulate intrusive thoughts may benefit from developing methods directed to find the right amount of suppression for each individual combined with techniques such as transcranial magnetic stimulation and cognitive behavioral interventions involving the training of non-judgmental awareness of the disturbing thoughts.

## Figures and Tables

**Figure 1 biomedicines-10-01555-f001:**
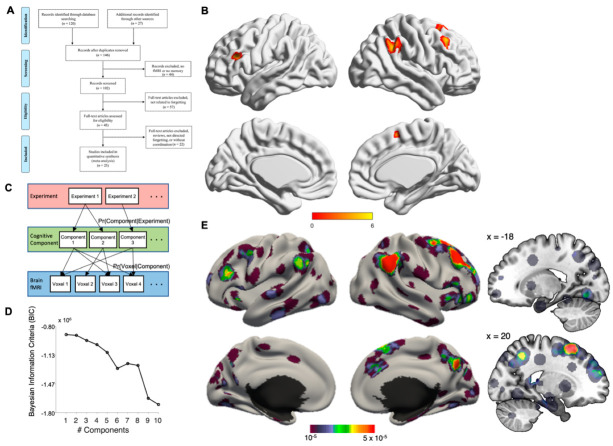
(**A**) PRISMA flow diagram; (**B**) results of ALE analysis; (**C**) Schematic diagram of LDA, (**D**) model selection of LDA results by BIC, and (**E**) LDA results (right hand side sagittal slides showing hippocampal & subcortical activations). Color bars at bottom of panel (**B**,**E**) represents the display threshold of the blobs presented.

**Figure 2 biomedicines-10-01555-f002:**
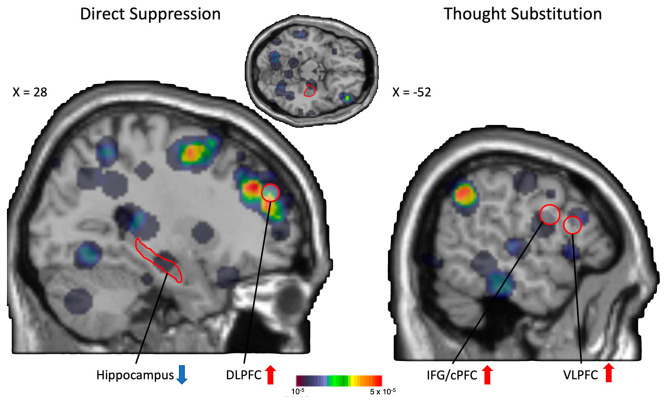
Distinct neural systems for direct suppression and thought substitution, as proposed by Benoit et al. (2012). They proposed that direct suppression involves recruitment of DLPFC and disengagement of hippocampus while thought substitution recruits IFG (caudal PFC) and VLPFC (ROIs in red enclosures, red upward arrows represent hypothetized engagement of brain regions whereas blue downward arrows represent hypothetized disengagement of brain regions). Results from our LDA analysis was overlaid on the MNI anatomical template. We observed distributed activities in all ROIs being mentioned. Here we treat direct suppression and inhibition are interchangeable constructs.

**Table 1 biomedicines-10-01555-t001:** List of fMRI studies included in the current meta-analysis.

Studies	n	Age	Software ^1^	Paradigm ^2^	Stimuli	Contrast
Anderson et al., 2004 [17]	24 (10F)	29–31	SPM99	T/NT	word pairs	suppression > recall
Bastin et al., 2012 [52]	17 (8F)	20–32	SPM5	DF	6-letter words	To be forget-forget > To be remember-forget
Benoit et al., 2012 [37]	18 (12F)	23.7	SPM8	T/NT	word pairs	suppression > recall
Benoit et al., 2015 [53]	16 (8F)	22	SPM8	T/NT	Picture	suppression > recall
Butler et al., 2010 [54]	14 (7F)	22.6	BV	T/NT	emotion pictures	NT > T (neutral)
Depue et al., 2007 [55]	16 (8F)	19–29	FSL	T/NT	face-picture pairs	Suppression > recall
Depue et al., 2016 [56]	21 (10F)	21.5	FSL	T/NT	neutral face pictures	Suppression > recall
Gagnepain et al., 2014 [57]	24 (11F)	22.	SPM8	T/NT	word-object pairs	Suppression > recall
Gagnepain et al., 2017 [58]	22 (8F)	18–35	SPM12	T/NT	face-scene pairs	NT > T
Gamboa et al., 2018 [28]	31 (15F)	27.5	SPM12	DF	vocal words	To be Forget > to be remember
Hanslmayr et al., 2012 [59]	22 (15F)	23.05	SPM5	DF	words	To be Forget > to be remember
Marchewka et al., 2016 [60]	18 (18F)	22.02	SPM12	DF	emotional pictures	TBF-F > TBR-F
Noreen et al., 2016 [38]	22 (18F)	18–29	SPM8	T/NT	word-autobiographic-memory pairs	no-think > think
Nowicka et al., 2011 [27]	16 (8F)	26.6	SPM8	DF	emotional pictures	TBF > TBR for neutral pictures
Reber et al., 2002 [61]	12 (9F)	20	NA	DF	faces	TBF > TBR
Rizio et al., 2013 [62]	24 (NA)	21.11	SPM8	DF	words	TBF > TBR
Sacchet et al., 2017 [63]	16 (8F)	31.7	AFNI	T/NT	word-pairs	no-think > think
Wang et al., 2019 [64]	20 (10F)	23.6	SPM 12	DF	pictures (scene, faces, objects)	TBF > TBR
Wierzba et al., 2018 [29]	24 (24F)	24.6	SPM12	DF	neutral/affective words	TBF > TBR
Wylie et al., 2008 [35]	11 (6F)	26	AFNI	DF	word pairs	TBF > TBR
Yang, T. et al., 2016 [30]	21 (13F)	22.19	SPM8	DF	word pairs	TBF > TBR (neutral words)
Yang, W. et al., 2013 [31]	25 (14F)	30	SPM8	DF	word pairs	TBF > TBR (neutral words)
Yang, W. et al., 2016 [65]	32 (10F)	30	SPM8	DF	word pairs	TBF > TBR

^1^ SPM = Statistical Parametric Mapping (The Wellcome Center for Human Neuroimaging, UCL Queens Square Institute of Neurology, London, UK); BV = Brain Voyager (Brain Innovation, Inc., The Netherlands); FSL = FMIRB Software Library (FMRIB, Oxford, UK); AFNI = Analysis of Functional NeuroImages (National Institute of Mental Health, USA); ^2^ DF = Directed forgetting; T/NT = think/no-think; TBF = To be forget; TBR = To be remember; TBF-F = To be forget and forget; TBR-F = To be remember but forget; NT > T = No think > Think.

**Table 2 biomedicines-10-01555-t002:** ALE and LDA results.

	Coordinates (MNI)					
Cluster	X	Y	Z	Number of Voxels	L/R	Anatomical Structure
**ALE**						
1	16	16	60	221	R	Superior Frontal Gyrus
2	58	−46	36	212	R	Inferior Parietal Lobe
3	42	24	44	160	R	Middle Frontal Gyrus
4	−42	28	24	117	L	Middle Frontal Gyrus
**LDA**						
1	−45	15	1	4091	L	Inferior Frontal Gyrus
**2**	**−27**	**45**	**21**	**724**	**L**	**Middle Frontal Gyrus**
**3**	**37**	**27**	**41**	**16,556**	**R**	**Middle Frontal Gyrus**
4	−21	51	−3	536	L	Orbitofrontal Gyrus
5	−43	−1	45	2122	L	Precentral Gyrus
6	−15	−3	45	498	L	Middle Cingulum
7	21	−39	43	515	R	Middle Cingulum
8	57	−37	13	1104	R	Superior Temporal Gyrus
9	−55	−37	−17	2934	L	Middle Temporal Gyrus
10	69	−25	−17	1012	R	Middle Temporal Gyrus
11	53	−25	−33	1050	R	Inferior Temporal Gyrus
12	−11	−15	−23	552	L	Hippocampus
13	25	−25	−17	1500	R	Parahippocampal Gyrus
14	−55	−59	39	1182	L	Inferior Parietal Lobe
**15**	**57**	**47**	**41**	**13,717**	**R**	**Inferior Parietal Lobe**
16	−7	−39	63	536	L	Precuneus
17	−43	−79	−5	931	L	Inferior Occipital Gyrus
18	49	−87	−3	398	R	Infeiror Occipital Gyrus
19	−15	−75	−7	1332	L	Lingual Gyrus
20	25	−71	−9	4213	R	Lingual Gyrus
21	13	−93	3	1039	R	Calcarine Gyrus
22	−33	−59	−23	552	L	Cerebellum
23	27	−79	−35	498	R	Cerebellum

Note: LDA results in BOLD overlap with ALE activations.

## Data Availability

Documentations of our literature search, screening and exclusion can be found in https://osf.io/ph278/.

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
