# Peer review of "Obliviate! Reviewing Neural Fundamentals of Intentional Forgetting from a Meta-Analytic Perspective"

_biomedicines, 2022, doi:10.3390/biomedicines10071555_

Round 1

Reviewer 1 Report

Authors responded to all my concerns. I have no further comments. Thank you.

Author Response

Thank you very much for your comments all along

Reviewer 2 Report

The authors have added information requested in review, and the manuscript is acceptable for publication

Author Response

Thank you very much for your comments all along

This manuscript is a resubmission of an earlier submission. The following is a list of the peer review reports and author responses from that submission.

Round 1

Reviewer 1 Report

This paper is a meta-analysis aiming to map the brain regions involved in intentional forgetting (IF). The authors identified 23 studies up to May 2020, and used 2 complementary approaches: Activation Likelihood Estimation (ALE) and Latent Dirichlet Allocation (LDA). The authors found activation in a fronto-parietal network during IF using ALE, and a fronto-medial temporal lobe network using LDA.

This is an interesting study, well written. Below are my comments/questions.

Introduction:

Line 99: using “caudal prefrontal cortex” is quite unusual to refer to a human brain region to my knowledge, it typically rather refers to animal anatomy. Would it be possible to rename using more “typical” phrasing such as dorso lateral prefrontal cortex, etc?

Section 1.3: Authors should add references about the ALE and LDA approaches at that point since it is the first time they refer to these methods.

Method:

Selection of the study: Why has the selection of studies stopped in May 2020? It is almost 2 years ago. This should be updated with the most recent studies.

Please provide an exact age range for the selection of the populations. How is “mid-aged adults” defined?

I have used ALE in the past and I know that covariates can be entered. In this study, I don’t see any covariates tested, or am I wrong? The software (SPM vs FSL, etc), age,sex, etc should be added as covariates. This is very typical for other ALE meta-analyses.

Are all the 23 studies based on fMRI? It is not indicated in Table 1.

Line 321 “We decided to use LDA …”: I do not understand the justification.

Based on the results and the selection of the number of components for from the LDA approach, the BIC is really really high! (BIC=-80,000, based on Figure 1). While this value for 1 component is the closest to zero, this value may suggest that the data are not really appropriate for the model. It is not because it can be done that it is appropriate to do so. This may also partially explain the large differences between the 2 approaches (see my other comment in the next section) What are your thoughts about that? More discussion may be needed.

Results

Table 2: in the ALE result section, “Right” or “left” from the anatomical structure description should be deleted.

There are large differences between the ALE and LDA analyses in regards to the number of significant regions. Do you think it is caused by differences in statistical sensitivity or actual findings related to the thought process? This should be discussed.

Discussion:

I think Figure 2 is quite unclear, the arrow points to regions with no activation or “holes” (e.g., cPFC). Also the blue and red arrows are not explained in the legend. It seems that the cPFC and VLPFC are basically the same region? I think this figure would benefit to be redone with better display of the activated regions. Directly pointing to the activated regions might also benefit for the interpretation/explanation.

Reviewer 2 Report

General Comments and Suggestions

    The goals of this study were to quantify the neural correlates of intentional forgetting and, to the extent possible, determine the support for two current hypothesized models of intentional forgetting.  The choice of meta-analytic methods could be better justified, but over all this was an appropriate method to achieve the stated goals.  The results section is weak, with concerns and suggested methods of improving this section noted below.  The discussion section is generally sound, but becomes a bit unfocused  in the final paragraphs.  As the authors reach a conclusion, a clearer discussion of the logic of that conclusion would strengthen this section.  In addition, the authors use the term ‘network’ somewhat loosely.  A clearer definition of what they mean by this term is needed.  For example, when discussing the ‘core network’ the focus is on general concepts of the role of that region in attention and focus and not on its participation in any defined network. The methods section, particularly in regards to the process and results of DLA analysis, needs to be improved.  Specific concerns are noted below.

Specific Concerns:

  1. INTRODUCTION

This section is generally well-written and clearly indicates the need for such an analysis.  While information on the types of analysis method are provided elsewhere in the manuscript, this section should include a brief rationale for the decision to use two methods of meta-analysis (Section 1.3). In addition, the choice to limit analysis to young adults should be justified and clearly noted in the results of PRISMA analysis.

  1. RESULTS

While the methods section does provide information on inclusion/exclusion criteria, this information, together with the results of the PRISMA search process, should be reported in precis form in this section.

  • In presenting PRISMA search results, the authors note 120 records identified through “database search” and 17 identified via “other sources” to obtain a total of 146 records, which is mathematically incorrect – please correct this information.

This reviewer has significant concerns with the results presented in Figure 1 from the DLA analysis that may be somewhat ameliorated by a much clearer description of the actual parameterization of this analytic method.

  • The correspondence between this method and the MLE analysis in posterior partietal and DLPFC is strong. However, in the DLA analysis as presented in Figure 1, the signal in hippocampus, primary visual cortex and cerebellum is quite weak.  Rather, there appears to be signal in inferior temporo-occipial (fusiform area), inferior and middle temporal gyr, ventromedial/anterior cingulate area, and perhaps frontal operculum that are not seen in the MLE analysis.  A more careful and detailed analysis of these data is needed to clarify the choice of label for presented results of this analysis
  • This reviewer suggests choosing a parcellation scheme that will enable clearer ROI labeling and presentation of the matrix on which DLA factorization was accomplished to aid in label clarification and ground it in recognized labeling schema
  1. DISCUSSION

The introductory paragraph recapitulates the author’s assertion that ALE analysis returned a core network subserving intentional forgetting, while DLA returned a supportive network.  These are the author’s interpretation of results and they are not unreasonable, but they should be supported by presented data and the literature rather than being presented as ‘truth’.  In particular, the identification of these regions as both ‘supportive’ and as a ‘network’ requires more support than is provided in the current discussion (which, for example, never includes a discussion of the role of the cerebellum).  It would be helpful to discuss studies in which any or all of these regions were reported as functioning in IF, which is only done for the hippocampus.

In the paragraph (section 3.2) describing the ‘supportive network’, the authors discuss hippocampal and rMFG connectivity in the think-no-think task, and data indicating a function in retrieval suppression.  While this section is supportive of these regions as important in IF, and thus of their interpretation (at least in part as these do not represent all of the regions that are suggested as part of this ‘network’), they authors fail to make that connection explicit.

Section 3.3 is rather unfocused and does not clearly develop the final conclusion reached.  This is not due to a lack of information, rather it points to a need to tighten the argument being made.

METHODS

There is a mathematical error in the results concerning the number of initially identified records that is not resolved in the methods section where the search mechanisms are inadequately detailed.

  • An initial search utilized PubMed, but the number of identified articles based on noted search terms is not noted
  • The PRISMA method requires additional database searches and it is not clear whether these were indeed performed, nor what databases (e.g. Web of Science, IDIS, etc)
  • In line 273, the sentence fragment (“Additional studies were identified through literature by the author …”) is uninformative (see bullet above)

In describing the LDA method, the authors reference two previous uses of this model in neuroimaging studies – one by Yeo et al. (2015) and a second by Zhang et al (2016) but neither of these is cited in the reference section.

  • The authors should note that this method was initially proposed in graph network analyses by Zhang et al (IEEE Intell. Secur. Informatics (2007)doi:10.1109/ISI.2007.379553), and its advantage over standard ICA or clustering methods is that a given brain region is allowed to belong to multiple networks
  • This analysis is accomplished by factorizing a matrix, but there is no description of the matrix used or of the factorized networks resulting from this step and this information is important to include
  • The authors should also indicate their choice of model parameters for this analytic method
  • The authors provide an analysis, using BIC, to derive the number of components to be accepted but do not provide a threshold value based upon this analysis – doing so would likely aid in interpretation and display of presented results

Figures

Figure 2: please provide a legend indicative of the color values for identify ROIs. The red circle in this figure purporting to represent cPFC is centered on a sulcal space and the center point of the VLPFC is an intersection of low valued and non-significant cortex which does not support the statement that activity in these ROIs is observed in support of the thought substitution hypothesis

Minor Editorial Concerns

A few editorial/language concerns are noted below:

Section 1.2, line 84:  Typo in header: please change ‘or’ to ‘of’

Section3, line138-139: This sentence is grammatically weak and unclear.  Please re-word